# The Impact of the COVID-19 Pandemic on Postpartum Maternal Mental Health

**DOI:** 10.3390/jpm13010056

**Published:** 2022-12-27

**Authors:** Lavinia De Chiara, Gloria Angeletti, Gaia Anibaldi, Chiara Chetoni, Flavia Gualtieri, Francesca Forcina, Paride Bargagna, Georgios Demetrios Kotzalidis, Tommaso Callovini, Marco Bonito, Alexia Emilia Koukopoulos, Alessio Simonetti

**Affiliations:** 1Department of Neurosciences, Mental Health, and Sensory Organs (NESMOS), Sapienza University, Via di Grottarossa 1035-1039, 00189 Rome, Italy; 2Centre for Prevention and Treatment of Women’s Mental Health Problems at Sant’Andrea Hospital of Rome, Sapienza University, Via di Grottarossa 1035-1039, 00189 Rome, Italy; 3Centro Lucio Bini, ARETÆUS Onlus, Via Crescenzio 42, 00193 Rome, Italy; 4Department of Medicine and Surgery, University of Milano Bicocca, 20900 Monza, Italy; 5Dipartimento Materno Infantile, San Pietro Fatebenefratelli Hospital, 00189 Rome, Italy; 6Department of Human Neuroscience, Sapienza University, 00185 Rome, Italy; 7Menninger Department of Psychiatry and Behavioral Sciences, Baylor College of Medicine, Houston, TX 77030, USA; 8Department of Neurology and Psychiatry, Sapienza University of Rome, 00185 Rome, Italy

**Keywords:** COVID-19, postpartum mental health, anxiety, depression, mania, hypomania

## Abstract

Objectives: There are reports of mental health worsening during the COVID-19 pandemic. We aimed to assess whether this occurred in women who were pregnant at baseline (late 2019) and unaware of the pandemic, and who delivered after the implementation of COVID-19 restrictions and threat (March–April 2020). To compare the pandemic period with the pre-pandemic, we capitalized on a retrospective 2014–2015 perinatal sample which had had affective symptoms assessed. Methods: The COVID sample were administered the Postnatal Depression Scale (EPDS), Zung Self-Rating Anxiety Scale (SAS), Hypomania Checklist-32 (HCL-32), Pittsburgh Sleep Quality Index (PSQI), and Perceived Stress Scale (PSS) at T0 (pregnancy) and T1 (post-delivery). The Non-COVID sample had completed EPDS and HCL-32 at the same timepoints. Results: The COVID sample included 72 women, aged 21–46 years (mean = 33.25 years ± 4.69), and the Non-COVID sample included 68 perinatal women, aged 21–46 years (mean = 34.01 years ± 4.68). Our study showed greater levels of mild depression in T1 among the COVID sample compared to the Non-COVID sample. No significant differences in terms of major depression and suicidal ideation were found. The levels of hypomania were significantly different between the two groups at T1, with the COVID sample scoring higher than the Non-COVID sample. This may be related to the high levels of perceived stress we found during the postpartum evaluation in the COVID sample. Limitations: There was a relatively small sample size. Conclusions: New mothers responded to the pandemic with less mental health impairment than expected, differently from the general population. Women delivering amidst the pandemic did not differ in depressive and anxiety symptoms from their pre-pandemic scores and from pre-pandemic women. Because stress responses have high energy costs, it is optimal for maternal animals to minimize such high metabolic costs during motherhood. Evidence suggests that reproductive experience alters the female brain in adaptive ways. This maternal brain plasticity facilitates a higher purpose, the continuation of the species. This may point to the recruitment of motherhood-related resources, for potentially overcoming the effects of the pandemic on mental health.

## 1. Introduction

Coronavirus disease 2019 (COVID-19), caused by severe acute respiratory syndrome coronavirus 2 (SARS-CoV-2), was first detected in December 2019 in Wuhan (China) [1]. Since that moment, the disease has spread worldwide. Due to the growing case notification rates outside Chinese frontiers, on the 30 January 2020, the WHO declared a global health emergency [2]. In Italy, lockdown due to the Coronavirus Disease 2019 (COVID-19, also known as 2019-nCoV) health emergency started on 10 March and partially ended on 3 May 2020. The Italian population was placed in social isolation for almost 2 months, except for permissions regarding primary needs. Other countries, during these months, decided to enforce similar restrictive measures. There have been many hypotheses on the possible damage to mental health produced by the lockdown. Prati and Mancini [3] found the psychological impact of lockdown to be small and highly heterogeneous. Another study found that greater psychological flexibility mediated decreases in the adverse effects of trait anxiety on COVID-19 distress, anxiety, and depression. In particular, two psychological flexibility processes, observing unhelpful thoughts rather than taking them literally (defusion) and values-based action (committed action), mediated decreases in the negative effects of trait anxiety and mood on all mental health outcomes [4]. Another Italian study found that during lockdown, access to the emergency department, for persons with psychiatric symptoms, decreased by 56% compared to the previous two months [5]. However, such a reduction might also be secondary to reduced service accessibility, due to socioeconomic reasons [6]. Hence, lockdowns may not have adverse effects on mental health and during the lockdown the activation of resilience-related factors is possible, or other reasons may lie behind, and underpin, the observed differences.

At odds with the above, there is much evidence regarding a close correlation between the worsening of mental health and the lockdown in Italy. Anxiety, depression, and stress symptom levels were found to increase with time; in the last weeks of the lockdown, it showed increased severity [7]. Other findings also support that mental health worsened during lockdown [8,9].

It is important to identify those sub-populations that are most susceptible to mental health worsening after the pandemic lockdown. According to an Italian cross-sectional study, being female, aged less than 45 years, working from home or being underemployed were all identified as prominent risk factors for mental health worsening—particularly mood—because of the lockdown [10]. Many studies worldwide have also investigated the impact of the SARS-CoV-2 pandemic on mental health in the peripartum period and obtained evidence of increased incidences of anxiety and depression in pregnant women. A recent meta-analysis provides evidence that the COVID-19 pandemic significantly increased the risk of anxiety among women during pregnancy and perinatal period. [11]. Compared to the pre-pandemic period, in the COVID-19 pandemic period, pregnant women were found to be more likely to have thoughts of harming themselves [12] and higher prevalence rates of clinically relevant maternal depression and anxiety (higher than clinical diagnoses of anxiety and depressive disorders) [13]. These psychological changes were perceived as a result of the introduction of social distancing measures in new mothers 0–3 months post-delivery [13]. The isolation-induced worsening of postnatal wellbeing may be eased by social support during the COVID-19 pandemic outbreak [14]. However, other studies failed to detect mental health worsening during the peripartum period; Pariente et al. [15] found that delivering during the COVID-19 pandemic was independently associated with a lower risk of postpartum depression. An Italian survey reported the adoption of more maladaptive coping strategies in isolated women in northern Italy, where there was high-risk for COVID-19, while those living in less risky areas were at lower risk of adopting these; maladaptive strategies were associated with more depressive and anxiety symptoms [16]. Unfortunately, there are a lack of data on mania/hypomania and perinatal psychosis during the pandemic, with the bulk of the literature focusing on anxiety and depression during the pandemic. Most studies used the Edinburgh Postnatal Depression Scale (EPDS) to assess depressive symptoms. A retrospective chart-review study of perinatal women carried out in Ancona, Italy, during the timeframe from March 2020 to March 2021 and by administering the Edinburgh Postnatal Depression Scale (EPDS), the Fear of COVID-19 (FCV-19-S), and the Coronavirus Anxiety Scale (CAS), found that women, independent of previous psychiatric history, experienced increased levels of anxiety, fear, and psychological distress, due to subsequent isolation, quarantine, lockdown, and deprivation of their normal social support [17]. 

We undertook our study to resolve the inconsistencies in the literature, for example whether the pandemic outbreak was followed by increased perinatal psychopathology and which factors affected it. Additionally, we wanted to complete the lacking mood data by investigating the polar opposite to the depressive state: hypomania.

The aim of this study was to explore whether quarantine measures and the COVID-19 pandemic outbreak enhanced psychopathological distress in the immediate postpartum period. We investigated anhedonia, anxiety, and depression through the Edinburgh Postnatal Depression Scale (EPDS) [18] in the immediate postpartum period in an Italian perinatal population using the tool’s validated Italian version [19]. We also studied the presence of mania/hypomania and anxiety or panic. We compared the pandemic-related data with those of a retrospective control sample consisting of women who had given birth in 2015. We expected a greater increase in hypomanic symptoms than in depressive symptoms, in response to the increased stress imposed by the pandemic. This study would allow us to detect differences in the perinatal health status of mothers in Italy during the lockdown compared to pre-pandemic years. Very few studies in Italy have carried out such an extensive symptom screening and no one had a control sample analyzed with the same tests.

## 2. Materials and Methods

### 2.1. Research Setting

The study was developed in the context of a collaborative screening program including the Gynaecology and Obstetrics unit of San Pietro Fatebenefratelli Hospital of Rome, Italy, and the Centre for Prevention and Treatment of Women’s Mental Health Problems, Psychiatry Unit, Sapienza University, Faculty of Medicine and Psychology, at Sant’Andrea Hospital of Rome, Italy. It involved all consecutive women attending fetal monitoring at the above-mentioned unit during the periods July 2014–July 2015 and July 2019–July 2020.

Participants provided written informed consent, in agreement with all applicable regulatory and Good Clinical Practice guidelines, fully respecting the Ethical Principles for Medical Research Involving Human Subjects, as adopted by the 18th World Medical Association General Assembly (WMA GA), Helsinki, Finland, June 1964, and subsequently amended by the 64th WMA GA, Fortaleza, Brazil, October 2013. The study received approval from the local ethics committees (Board of the Sant’Andrea Hospital of Rome and San Pietro Fatebenefratelli Hospital of Rome) and was authorized through the Prot. N. 2471/CE Lazio1.

### 2.2. Participants

Recruited were 72 women, between July and September 2019, during their third trimester of pregnancy who were referred to the Gynaecology and Obstetrics service of San Pietro Fatebenefratelli Hospital before the COVID-19 pandemic outbreak. Consecutive women, who had consented to being contacted in the postnatal period, were called by two trained psychologists in our team 6 months following the birth of their baby, between April and July 2020 during the COVID-19 health emergency, and were invited to complete the questionnaires through an online system (Google Forms). We compared data of 68 pregnant women consecutively recruited from the same screening program collected between July and September 2014, with follow-ups between April and July 2015 (Figure 1).

Exclusion criteria were failure to provide free informed consent and insufficient comprehension of the Italian language that prevented participants from completing the questionnaires.

The final study sample included 140 pregnant women, aged 21–46 years (Mean = 33.62 ± (standard deviation, SD) 4.68), with 93.9% of participants (N = 138) being Italian. The majority held a university degree (N = 88, 62.9%) and were employed (N = 114, 81.4%). More descriptive statistics of the two subsamples are presented in the Results and Table 1.

### 2.3. Measures

Screening tools were administered by physicians and psychologists at the Centre for Prevention and Treatment of Women’s Mental Health Problems at Sant’Andrea Hospital of Rome.

Participants completed the following questionnaires:

**Perinatal Interview (PI)** is a paper-and-pencil questionnaire to collect sociodemographic and clinical information, allowing us to investigate predictive and protective factors for the development of psychiatric disorders. Besides birthdate and place, nationality, educational level, job, and marital status, the PI investigates the following: habits; voluptuary substance use (including tobacco and alcohol); physiological rhythms; past surgery; past and current pharmacological treatment; gynaecological and obstetric history, focusing on current and past pregnancies; possible presence of premenstrual syndrome; spontaneous or surgically-induced abortions; obstetric complications; means by which pregnancy has been obtained (spontaneous vs. medically-assisted reproduction [MAR]); past and current personal and family psychiatric history, and possible psychiatric treatments.

**Edinburgh Postnatal Depression Scale (EPDS)** [18] is a 10-item self-rated questionnaire to screen for the risk of depression, anxiety, and suicidal ideation during the peripartum period. Initially developed for the identification of postpartum depression [18], the EPDS was later validated for prenatal screening as well [20]. Thanks to its reliability and brevity, this easy to complete and interpret tool became a standard in perinatal care and is recommended by the National Institute for Health and Care Excellence guidelines [21], and cited among the main depression screening instruments by the American College of Obstetricians and Gynecologists [22]. The questionnaire refers to how the woman felt in the last seven days and each item is scored on a Likert-scale from 0 to 3 (variously labelled). Items 1 and 2 assess anhedonia, 3 guilt, 4 anxiety, 5 fear or panic, 6 helplessness, 7 sleep disorders, 8 sadness, 9 tendency to cry, and 10 tendency towards self-harm. Items 1, 2, and 4 are scored 0–3, all others 3–0 (reverse). Higher scores indicate more risk of depression. In the original English version, a cutoff between 12 and 13 showed 86% sensitivity and 78% specificity; however, the authors suggested a threshold between 9 and 10 for community screening [18]. This cutoff has been endorsed by others [23,24]. Italian validation studies identified 9–10 [25] and 12–13 [19] as optimal cutoffs. Furthermore, the combined score on items 3, 4, and 5 has been termed EPDS-3A and assumed as a proxy for the screening for anxiety disorders, with a ≥6 cutoff postpartum [26] and ≥4 antenatally [27]. Here, we adopted the latter cutoff for risk of anxiety. In the original study, authors recommend to immediately look for the score on item 10 (self-harm) and refer the patient for further evaluation in case the score is different from 0. We followed this suggestion strictly. In this study, we adopted the following cutoffs: total EPDS ≥ 12 = “risk for major depression”, total EPDS ≥ 9 = “risk for mild depression”, score on item 10 > 0 = “suicide ideation”.

**PSS (The Perceived Stress Scale),** by the American Sociological Association [28]. The PSS-10 is a shorter version of the original PSS-14. The PSS is the most widely used psychological instrument for measuring the perception of stress. Each item is rated on a 5-point Likert-type scale, ranging from 0  =  ‘never’ to 4  =  ‘very often’. It is a measure of the degree to which situations in one’s life are appraised as stressful. The internal consistency reliability of the PSS was examined by Cronbach’s alpha and the reasonable acceptability criterion which is ≥0.70. The PSS-10 comprises six negative (items 1, 2, 3, 8, 11, and 14) and four positive items (items 6, 7, 9, and 10). The total score of PSS is obtained by reversing the scores on the positive items and then summing across all the items, with a higher score indicating higher perceived stress. Possible total scores for PSS-10 range from 0 to 40. Low perceived stress: PSS-10 score <14; moderate perceived stress: PSS-10 score 14–26; high perceived stress: PSS score 27–40. We used the Italian version of the 10-item Perceived Stress Scale (PSS-10; Cronbach’s alpha = 0.74), translated and standardized by Mondo et al. [29].

**The Hypomania CheckList-32** (HCL-32) [30] is a 32 item self-rating questionnaire investigating lifetime history of hypomanic symptoms. Individuals scoring ≥ 14 are potentially with bipolar disorder/diathesis and should be carefully interviewed. The ideal cut-off point of the Italian version was 12 with a sensitivity of 0.85 and a specificity of 0.61 [31].

**The Pittsburgh Sleep Quality Index** (PSQI) [32,33] is a retrospective self-report questionnaire, that measures sleep quality and disturbances over the previous month. The PSQI assesses seven clinically derived components of subjective sleep quality: 1. sleep quality, 2. sleep latency, 3. sleep duration, 4. habitual sleep efficiency, 5. sleep disturbance, 6. use of sleep medications, and 7. daytime dysfunction. The PSQI yields a global score that represents the sum of the seven components scores, which are rated on a 4-point Likert scale ranging from 0 to 3, where 3 reflects the negative extreme of the Likert scale. A global score of 5 or higher is considered as an indicator of relevant sleep disturbances in at least two components, or of moderate difficulties in more than three components, discriminating between “good” and “bad” sleepers. In the Italian validation study [33], the PSQI showed high internal consistency with a Cronbach’s alpha of 0.84.

**The Zung Self-Rating Anxiety Scale** (SAS) [34] is a 20-item self-report assessment tool built to measure state anxiety levels. Raw scores range from 20 to 80. The initial cutoff was 50 [35], but the best cutoff was later proposed to be 40 for clinical settings and 36 for screening purposes [36]. The instrument is suited to investigate anxiety disorders [37] and showed strong correlations with other similar instruments [38].

### 2.4. Statistical Analysis

The sample was split into a COVID and a Non-COVID group. After controlling for normality of distribution with the Shapiro–Wilk test [39], continuous variables were summarized with mean (average) and standard-deviation. Categorical variables were summarized as absolute and percentage values. Between-group differences in sociodemographic characteristics were assessed at T0. Sociodemographic continuous variables (age) have been investigated through t-tests. Categorical variables (nationality, educational level, professional status, active medical condition, medical treatment, psychiatric history, previous psychiatric drug treatment, smokes during pregnancy, alcohol use during pregnancy, overuse of coffee, tea or energy drinks (>3 cup/die), use of drugs during pregnancy, premenstrual syndrome, psychiatric family history, past abortions, primiparity, medically-assisted reproduction for current pregnancy, previous perinatal psychiatric episodes, pregnancy complications) have been investigated through Chi-square tests. Cut-off for clinical significance was set at *p* < 0.05. For all analyses we used the IBM Statistical Package for the Social Sciences software, version 25 (IBM SPSS 25, Armonk, NY, USA, 2017). The differences between groups regarding levels of depression, anxiety, and hypomanic symptoms and their changes through time have been investigated with repeated-measures general linear models (GLM). In each GLM, group (COVID, Non-COVID) was the independent variable, whereas HCL-32 total scores, number of subjects fulfilling the criteria of risk of major depression (EPDS > 12), mild depression (EPDS > 9), anxiety disorder (EPDS3A > 4), suicidal ideation (EPDS-item 10 > 0) were dependent variables. Time effect (T0 and T1) was investigated as a within-subject variable. Time by group interaction effect was additionally investigated. Multiple comparison correction was applied for scores regarding EPDS. Post hoc t-tests were used in the case of a main effect or in the case of group by time significant interaction effect. Exclusively in COVID women, time changes in levels of stress, anxiety, and insomnia were assessed with multiple t-tests. Specifically, time (T0, T1) was an independent variable, whereas SAS, PSQI, and PSS were dependent variables. 

## 3. Results

### 3.1. Descriptive Statistics

The samples are homogeneous in terms of socio-demographic characteristics, as demonstrated in Table 1, where descriptive statistics are shown, including all characteristics considered in both samples.

### 3.2. Non-COVID Sample

The Non-COVID sample included 68 Italian-fluent adult pregnant women from the general population, screened once during their third trimester of pregnancy (T0) and those who agreed to follow-up were tested again at six months postpartum (T1). Participants with an incomplete EPDS were excluded from the final analysis. Participants were aged 21–46 years (*mean* = 34.01 years ± 4.68).

### 3.3. COVID Sample 

The COVID sample included 72 pregnant women, screened once during their third trimester of pregnancy (T0) and those who agreed to follow-up were tested again at six months postpartum (T1). Participants were aged 21–46 years (*mean* = 33.25 years ± 4.69).

Of the COVID sample, 10 (14.1%) women were found to be affected by postpartum major depression and 62 (85.9%) were not affected by postpartum depression at the EPDS at T1, by using a cutoff of 12—the most commonly employed in the literature [40,41]. A cut-off of 9 has been suggested [25] to reduce the proportion of false negatives. According to this cutoff, the proportion of women affected by postpartum Mild Depression on the EPDS at T1 was higher than at T0, and equal to 21 (29.6%) (Table 2).

Anxiety symptoms were detected in 12 (16.9%) women, both through the EPDS-3A and the SAS (Table 3).

Twenty-one participants (29.6%) scored positive on the HCL-32, 16 (21.3%) scored negative. Twenty-six (36.6%) women were positive on the PSQI (Table 4). The mean score of the PSS was 15.51 ± 7.698. Scores obtained on the administered rating scales are shown in Table 4.

### 3.4. Comparisons between COVID and Non-COVID Samples

The COVID sample scored more than the Non-COVID at T1 on EPDS TOT (6.08 ± 4.494 vs. 4.85 ± 3.391) and EPDS ANX (2.96 ± 2.148 vs. 2.56 ± 1.757), but these differences were not statistically significant.

Significant difference was found between COVID and Non-COVID samples, in terms of proportion of depressed/non-depressed women based on the cutoff of 9 for Mild Depression (Table 2).

Statistically significant differences were found between Non-COVID and COVID samples on HCL-32 at T1 [F(6.743) = 201,147; *p* = 0.010]. The COVID sample scored 10.96 ± 4.483, while Non-COVID scored 8.52 ± 6.362.

We found significant a difference in the severity of depressive and anxiety scores on the EPDS between T0 and T1 in the COVID sample (5.85 ± 4.051 vs. 6.08 ± 4.494. *p* = < 0.001 and 2.96 ± 2.148 vs. 3.11 ± 1.835 *p* = 0.024) (Table 4).

In regards to both the presence of major and mild depression, GLM showed the main differences in the groups. In both cases, the COVID group showed greater levels of either major or mild depression than the Non-COVID group. Exploratory post hoc t-tests were performed in order to investigate within which timeframe (T0, T1) this main difference is driven. Regarding the presence of major depression, differences between groups were present at T0, whereas at T1 they reached only a trend level of significance. Regarding the presence of mild depression, differences between groups were mainly driven by differences at T1, whereas differences at T0 were not significant. 

With regards to hypomanic symptoms, GLMs revealed differences in the groups, with COVID showing greater hypomanic symptoms than Non-COVID. Exploratory t-tests reealed that such a difference is mainly drivven by differences at T1, whereas differences at T0 were not significant. 

Regarding levels of stress, COVID showed worsening of levels of stress at T1, as demonstrated by an increase in PSS total scores. Regarding sleep quality, COVID showed an improvement in sleep quality over time, as demonstrated by a significant increase in PSQI total scores. 

We carried-out correlations using the Pearson’s r coefficient between all used instruments. Scores did not correlate with the mothers’ age. Besides the expected correlations between EPDS ANX and SAS scores, EPDS total depression and anxiety correlated with sleep disturbance and perceived stress levels. Hypomania scores did not correlate with any other scale.

## 4. Discussion

Our study showed greater levels of mild depression during the six-month postpartum period among women delivering during the COVID-19 pandemic, compared to the pre-pandemic period. This is in line with reports of the COVID-19 pandemic having negatively affected mental health in the general population. No significant differences in terms of major depression and suicidal ideation were found. There are many possible explanations for this phenomenon. For example, women who gave birth during the pandemic period may have received greater support from their family and partner during a period of risk for developing significant psychopathological symptoms [42]. So, it is possible that social support set off the negative effects of the pandemic and the related restrictions. In Italy, two-thirds of parents spent more time with their families during the lockdown than in previous periods [43]. It is important to remember that in the literature, the association between depressive symptoms and poor quality in relationships is 18–35% [44]. A close maternal–fetal attachment is likely to buffer postpartum anxiety symptoms, partially mediated through postpartum bonding and satisfaction with their partner. Therefore, strengthening maternal–fetal attachment and partnership during pregnancy has the potential to reduce maternal postpartum anxiety symptoms [45]. Spending time indoors with children during lockdown is recognized as a protective factor for mental health [46]. Additionally, family support proved to be an important protective factor for mental health during the lockdown [47].

In Italy there are fewer worries about renting a house or running out of money due to social cohesion, which was found to reduce the risk of depression and perinatal anxiety, and to be a protective factor for mental health during quarantine [48]. Resilience from mental health damage, from a catastrophic event of any origin, has been shown in pregnant and postpartum women and is linked to having a partner [49,50].

Regarding hospitalization for childbirth, no differences were found in the literature between women who gave birth before and during the pandemic period. This could be due to several factors, i.e., shorter duration of hospitalization, more in-depth clinical follow-ups, and thorough COVID-19 testing in hospitalized prepartum women [51]. For these reasons, the women included in this study were followed up in Rome, during a period in which most confirmed COVID-19 cases occurred in northern Italy. For example, women reporting fear of having their own child infected, reported higher depression levels compared to women without such fear [52]. Women who spent isolation in northern Italy—which was hit harder by the virus in terms of number of infected people, deaths, and media climate of perceived danger—were found to be at greater risk of symptoms of postpartum depression and perceived stress, compared to women who spent isolation in central or southern parts of the country [16]. Having had an acquaintance infected by the virus increased self-reported depression. Having been in touch with a COVID-19 case or with a symptomatic case might have increased the fear of infecting the child, and could have led new mothers to avoid contact and follow all the preventive measures, thus influencing the quality of maternal care. Finally, many disaster victims do not develop long- or short-term psychopathology thanks to high levels of resilience [53].

Women who went through postpartum during lockdown (COVID sample) had already higher anxiety levels during pregnancy (EPDS ANX T0) than the control Non-COVID sample. This is hard to explain, as neither sample knew about COVID-19. We may suppose that the most recent sample suffered a higher general stress period, but there are no data to support it. However, these levels decreased significantly in the postpartum (T1) and such a difference was no longer significant when comparing the two groups at T1 (Table 2 and Table 3).

As we expected, there was a greater increase in hypomanic symptoms than in depressive symptoms in response to the increased stress imposed by the pandemic. The levels of hypomania, as assessed through the HCL-32, were significantly different between the two groups at T1 (Table 3), with the COVID sample scoring higher than the Non-COVID sample. This may be related to the high levels of perceived stress we found during the postpartum evaluation in the COVID sample (PSS T1). We will not speculate further, as the PSS was administered only to the COVID sample. In emergency situations, stress, in association with hyperactivity and insomnia, can trigger hypomania. Furthermore, hypomania has been described as a defense mechanism against depression. Hypomania, in this perspective, is expressed as the denial of the underlying powerful depressive dynamics, and results in the presence of hyperactivation and a state of elevated mood [54]. However, the lack of correlation between HCL-32 scores and scores on other scales prevents us from making speculations.

Our data collection was made in the first months of the pandemic; hence, we could hypothesize that the prolongation of the stress could have led to different values later. Most reviewed studies on the impact of the quarantine in different situations reported negative psychological effects, including post-traumatic stress symptoms [55] and an important stressor included longer duration [56]. For this reason, further studies are needed to assess the impact of continued stress.

It is noteworthy that PSS scores at T1 in the COVID sample increased from 11.50 ± 5.795 at T0 to 15.51 ± 7.698 at T1 (although this was not statistically significant) during the COVID-19 health emergency and the quarantine, which started in March 2020. Increased level of stress perception is not unexpected. The COVID-19 pandemic is an uncontrollable stressor which changed daily life and had a huge impact on society, with health and economic repercussions. The COVID-19 pandemic represents a severely stressful event with widespread disruptions, including loss of income and housing, social/physical distancing, and fearfulness about infection, that may contribute to changes in mental health symptoms. Cross-sectional COVID-19 pandemic studies [57,58] suggest that similar to prior natural disaster research [50,59,60,61,62,63], adult mental health symptoms are elevated during the pandemic [64]. This could be explained both by fear of disease and its consequences, but also by uncertainty about its course. Furthermore, the role of the media in providing information should not be underestimated. Many people cited poor information from public health authorities as a stressor, reporting insufficient clear guidelines about action to take, and confusion about the purpose of the quarantine; the lack of clarity about the different levels of risk led people to fear the worst [56].

Limitations and Strengths. The main limitation of our study is the reduced sample size. Furthermore, COVID sample T0 scores were greater than Non-COVID sample T0 scores, already at the start point. We explored only partly the mental health well-being of new mothers, as we did not assess PTSD, impulsiveness, and other areas of symptomatology. However, we explored usually unexplored areas of perinatal mental health. However, at a screening during the COVID-19 pandemic, it is challenging to make participants adhere to complicated protocols during the perinatal period. Additionally, there is a need to avoid time-consuming procedures. Hence, we chose the validated and widely used EPDS along with the HCL-32 and the PSS, which fitted our purposes. A strength of this study is that the baseline population had no selection bias. Our study’s ethicality consists of identifying persons suspected to have either depression or anxiety, taking care of them by our service, and further supporting them during their postpartum period. Another limitation was the low proportion of patients responding to follow-up; setting changes might have been responsible for such attrition. Furthermore, we did not apply the strict statistical cutoff of the Bonferroni correction, due to the exploratory nature of our study. Strengths include face-to-face interviews (when surveyed at T1, women were called by doctors who they were acquainted with) and a retrospective sample during a time when the pandemic was not envisaged. Furthermore, most studies focus on anxious and depressive symptoms only, while we also investigated hypomania.

Future Implication. Evidence suggests that the reproductive experience alters the female brain in adaptive ways. Among these enhancements are marked and significant behavioral modifications, due to reported changes in the brains of maternal rats, which include increases in spatial memory, aggression, and exploratory behavior, as well as decreased anxiety, and attenuated stress responsiveness. Indeed, pregnancy and motherhood are well documented periods when there is a reduction in response to stressors [65,66,67]. Because stress responses have high energy costs, it is optimal for maternal animals’ needs to minimize such high metabolic costs during lactation.

During the perinatal period, the salience network is activated in response to the most relevant stimuli, resulting in a state of alert being fundamental, among others, for threat detection. Maternal concerns focus on the well-being of the infant where vigilant protectiveness and harm-avoidant behaviors are essential. This network is built around paralimbic structures—particularly the dorsal anterior cingulate and orbital fronto-insular cortices—and has a strong connectivity to subcortical and limbic structures [68]. A common challenge for parents is to preserve their own regulated emotional state in front of threats, or while caring for their immature and deregulated child.

Future studies should focus on maternal strengths and positive parenting practices among mothers experiencing adversity. Chan ASW and colleagues, in their review, suggested that the health care system should offer coaching and instruction on psychosocial problems to healthcare service administrators, emergency personnel, and health care providers [69]. They recommended that mental health and emergency response systems could work together to identify, establish, and allocate evidence-based resources such as disaster-related mental health, psychological well-being crisis and referral, special patient needs, and alarm and distress treatment. We suggest implementing consultation initiatives for new mothers to recognize their psychosocial needs, and to provide them with therapeutic services and social initiatives that should be integrated into overall pandemic healthcare.

## 5. Conclusions

New mothers responded to the pandemic with less mental health impairment than expected, differently from the general population. Women delivering amidst the pandemic did not differ in their depressive and anxiety symptoms from their pre-pandemic scores and from pre-pandemic women. As we expected there was a greater level of hypomanic symptoms during the COVID period, in response to the increased stress imposed by the pandemic. Taken together, these considerations and our data on maternal mental health during the COVID-19 health emergency seem to point to the ability of humans to face adversity, so as to safeguard the species. These stress factors seem not to be strong enough to impair the normal processes of maternal care and emotional stability; on the other hand, they could actually trigger defense mechanisms that result in better performance.

## Figures and Tables

**Figure 1 jpm-13-00056-f001:**
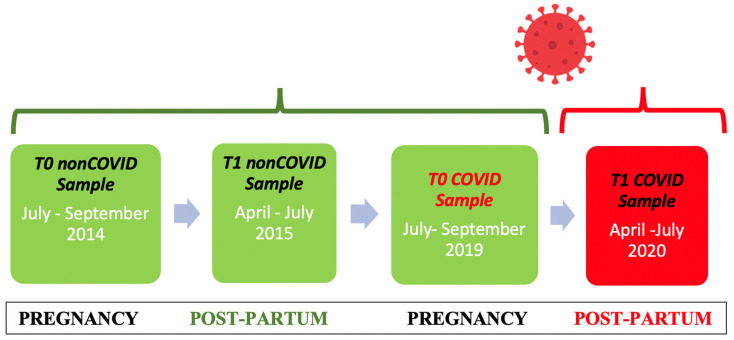
Design of the study.

**Table 1 jpm-13-00056-t001:** Socio-demographic and clinical characteristics of the COVID (N = 72) and Non-COVID (N = 68) samples (*p* values reflect Student’s *t* for means and SDs, and χ^2^ for N and %).

Parameter	Total Group	COVID	Non-COVID	*p* COVID vs. Non-COVID
**Age, in years ( ± SD)**	33.62 ± 4.68	33.25 ± 4.69	34.01 ± 4.68	0.337
**Nationality**				0.495
Italian N (%)	131 (93.6)	66 (91.7)	65 (95.6)	
Other N (%)	9 (6.4)	6 (8.3)	3 (4.4)	
**Educational level**				0.967
Primary, N (%)	0	0	0	
Middle school, N (%)	4 (2.8)	2 (2.8)	2 (2.9)	
High school, N (%)	48 (34.3)	24 (33.3)	24 (35.3)	
College/ University, N (%)	88 (62.9)	46 (63.9)	42 (61.8)	
**Professional status**				1.000
Employed, N (%)	114 (81.4)	59 (81.9)	55 (80.9)	
Unemployed, N (%)	26 (18.6)	13 (18.1)	13 (19.1)	
**Active medical condition**				0.152
Yes N (%)	30 (21.4)	19 (26.4)	11 (83.8)	
No N (%)	109 (77.9)	52 (72.2)	57 (16.2)	
Missing	1 (0.7)	1 (1.4)	0	
**Medical treatment**				0.819
Yes N (%)	22 (15.7)	12 (16.7)	10 (14.7)	
No N (%)	118 (84.3)	60 (83.3)	58 (85.3)	
**Psychiatric history**				0.840
Positive N (%)	31 (22.1)	15 (20.8)	16 (23.5)	
Negative N (%)	106 (75.7)	54 (75)	52 (76.5)	
Missing	3 (2.1)	3 (4.2)	0	
**Past psychiatric drug treatment**				0.741
Yes N (%)	9 (6.4)	4 (5.6)	5 (7.4)	
No N (%)	128 (91.4)	66 (91.7)	62 (91.2)	
Missing	3 (2.1)	2 (2.8)	1 (1.5)	
**Psychiatric drug treatment during current pregnancy**				.
Yes N (%)	0	0	0	
No N (%)	140 (100)	72 (100)	68 (100)	
**Smokes**				0.116
Yes N (%)	7 (5)	6 (8.6)	1 (1.5)	
No N (%)	131 (93.6)	64 (91.4)	67 (98.5)	
Missing	2 (1.4)	2 (2.8)		
**Uses alcohol**				.
Yes N (%)	0	0	0	
No N (%)	140 (100)	72 (100)	68 (100)	
**Coffee, Tea or Energy drinks**				< 0.001
Yes N (%)	54 (38.6)	38 (52.8)	52 (76.5)	
No N (%)	86 (61.4)	34 (47.2)	15 (22.1)	
**Abuses of drugs**				0.497
Yes N (%)	2 (1.4)	2 (2.8)	0	
No N (%)	138 (98.6)	70 (97.2)	68 (100)	
**Premenstrual syndrome**				0.305
Yes N (%)	60 (42.9)	34 (47.2)	26 (38.2)	
No N (%)	79 (56.4)	37 (51.4)	42 (61.8)	
Missing	1 (0.7)	1 (1.4)	0	
**Psychiatric family history**				1.000
Positive N (%)	43 (30.7)	22 (30.6)	21 (30.9)	
Negative N (%)	93 (66.4)	47 (65.3)	46 (67.6)	
Missing	4 (2.9)	3 (4.2)	1 (1.5)	
Past abortions				0.591
Yes N (%)	47 (33.6)	26 (36.1)	21 (30.9)	
No N (%)	92 (65.7)	45 (62.5)	47 (69.1)	
Missing	1 (0.7)	1 (1.4)	0	
**Primiparas**				0.726
Yes N (%)	87 (62.1)	43 (59.7)	44 (64.7)	
No N (%)	52 (37.1)	28 (38.9)	24 (35.3)	
Missing	1 (0.7)	1 (1.4)	0	
**Medically-Assisted Reproduction**				0.495
Yes N (%)	9 (6.4)	6 (8.3)	3 (4.4)	
No N (%)	131 (93.6)	66 (91.7)	65 (95.6)	
**Previous Perinatal psychiatric episodes**				0.392
Yes N (%)	6 (11.3)	2 (6.9)	4 (16.6)	
No N (%)	47 (88.7)	27 (93.1)	20 (83.3)	
**Pregnancy Complications**				0.447
Yes N (%)	38 (27.1)	22 (30.6)	16 (23.5)	
No N (%)	99 (70.7)	49 (68.1)	50 (73.5)	
Missing	3 (2.1)	1 (1.4)	2 (2.9)	

**Table 2 jpm-13-00056-t002:** Clinical measures: Chi-Square comparisons between COVID and Non-COVID samples.

	COVID N (%)	Non-COVID N (%)	*p*-Value
**T0**			
**EPDS T0 ≥ 12**	7 (9.7)	1 (1.5)	0.063
**EPDS T0 ≥ 9**	18 (25)	8 (11.8)	0.052
**EPDS-3A T0 ≥ 4**	31 (43)	16 (23.5)	0.020
**Item 10 >0 T0**	2 (2.8)	0	0.497
**HCL 32 T0 ≥ 14**	25 (40.3)	24 (38.1)	0.856
**PSQI T0 ≥ 5**	51 (72.9)	21 (72.4)	1,000
**SAS T0 ≥ 36**	22 (31.9)	15 (51.7)	0.073
**T1**			
**EPDS T1 ≥12**	10 (14.1)	3 (4.4)	0.078
**EPDS T1 ≥9**	21 (29.6)	7 (10.3)	**0.006**
**EPDS-3A T1 ≥4**	12 (16.9)	4 (5.9)	0.061
**Item 10 >0 T1**	3 (4.2)	1 (1.5)	0.620
**HCL 32 T1 ≥ 14**	21 (29.6)	14 (21.5)	0.329
**PSQI T1 ≥ 5**	26 (36.1)		
**SAS T1 ≥ 36**	12 (16.7)		

Significant results in **bold**, *Abbreviations:* EPDS, Edinburgh Postnatal Depression Scale; EPDS-3A, Anxiety component of the Edinburgh Postnatal Depression Scale; HCL-32, Hypomania Checklist-32 items; PSQI, Pittsburgh Sleep Quality Index; PSS, Perceived Stress Scale; SAS, Zung Self-Rating Anxiety Scale.

**Table 3 jpm-13-00056-t003:** ANOVA one-way for EPDS, HCL-32, by COVID and Non-COVID samples.

		Sum of Squares	df	Mean Square	F	Sig.
**T0 EPDS TOTAL**	**Between Groups**	66,275	1	66,275	5583	**0.020**
	**Within Groups**	1,638,261	138	11,871		
	**Total**	1,704,536	139			
**T0 EPDS-3A**	**Between Groups**	20,102	1	20,102	6994	**0.009**
	**Within Groups**	396,641	138	2874		
	**Total**	416,743	139			
**T0 HCL-32 TOTAL**	**Between Groups**	68,354	1	68,354	1619	0.206
	**Within Groups**	5,192,318	123	42,214		
	**Total**	5,260,672	124			
**T1 EPDS TOTAL**	**Between Groups**	52,683	1	52,683	3305	0.071
	**Within Groups**	2,184,022	137	15,942		
	**Total**	2,236,705	138			
**T1 EPDS-3A**	**Between Groups**	5528	1	5528	1430	0.234
	**Within Groups**	529,638	137	3866		
	**Total**	535,165	138			
**T1 HCL 32 TOTAL**	**Between Groups**	201,147	1	201,147	6743	**0.010**
	**Within Groups**	3,997,089	134	29,829		
	**Total**	4,198,235	135			

Significant results in **bold**, *Abbreviations:* EPDS, Edinburgh Postnatal Depression Scale; EPDS-3A, Anxiety component of the Edinburgh Postnatal Depression Scale; HCL-32, Hypomania Checklist-32 items; PSQI, Pittsburgh Sleep Quality Index; PSS, Perceived Stress Scale; SAS, Zung Self-Rating Anxiety Scale; TOTAL, total score.

**Table 4 jpm-13-00056-t004:** Scores on psychometric scales in the Non-COVID and COVID samples. First three rows: EPDS, HCL 32, total scores at timepoints T0 and T1 in the Non-COVID sample. Last six rows: EPDS, HCL 32, PSS, PSQI, and SAS total scores at timepoints T0 and T1 in the COVID sample.

	T0	T1		Sum of Squares	df	Mean Square	F	*p*
**NON COVID**								
**EPDS TOTAL**	4.47 ± 2.657	4.85 ± 3.391	**Between Groups**	143,266	13	11,020	1805	0.066
			**Within Groups**	329,675	54	6105		
			**Total**	472,941	67			
**EPDS-3A**	2.35 ± 1.533	2.56 ± 1.757	**Between Groups**	56,114	7	8016	4743	**<0.001**
			**Within Groups**	101,415	60	1690		
			**Total**	157,529	67			
**HCL 32 TOTAL**	10.70 ± 6.838	8.52 ± 6.362	**Between Groups**	1,301,200	22	59,145	1520	0.128
			**Within Groups**	1439,783	37	38,913		
			**Total**	2,740,983	59			
**COVID**								
**EPDS TOTAL**	5.85 ± 4.051	6.08 ± 4.494	**Between Groups**	602,208	16	37,638	3614	**<0.001**
			**Within Groups**	562,383	54	10,415		
			**Total**	1,164,592	70			
**EPDS-3A**	3.11 ± 1.835	2.96 ± 2.148	**Between Groups**	57,101	8	7138	2432	**0.024**
			**Within Groups**	181,998	62	2935		
			**Total**	239,099	70			
**HCL 32 TOTAL**	12.18 ± 6.131	10.96 ± 4.483	**Between Groups**	794,903	19	41,837	1163	0.332
			**Within Groups**	1,474,507	41	35,964		
			**Total**	2,269,410	60			
**PSS TOTAL**	11.50 ± 5.795	15.51 ± 7.698	**Between Groups**	639,724	24	26,655	672	0.841
			**Within Groups**	1,268,417	32	39,638		
			**Total**	1,908,140	56			
**PSQI GLOBAL**	6.74 ± 3.369	4.38 ± 3.040	**Between Groups**	165,501	10	16,550	1558	0.143
			**Within Groups**	616,267	58	10,625		
			**Total**	781,768	68			
**SAS TOTAL**	33.78 ± 6.135	32.70 ± 4.451	**Between Groups**	982,774	16	61,423	2016	**0.030**
			**Within Groups**	1,553,756	51	30,466		
			**Total**	2,536,529	67			

Significant results in **bold**, *Abbreviations:* EPDS, Edinburgh Postnatal Depression Scale; EPDS-3A, Anxiety component of the Edinburgh Postnatal Depression Scale; HCL-32, Hypomania Checklist-32 items; PSQI, Pittsburgh Sleep Quality Index; PSS, Perceived Stress Scale; SAS, Zung Self-Rating Anxiety Scale; TOTAL, total score.

## Data Availability

A file with participants’ data anonymized will be provided to those making a reasonable request to the corresponding author.

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
