# Peer review of "The Impact of the COVID-19 Pandemic on Postpartum Maternal Mental Health"

_jpm, 2022, doi:10.3390/jpm13010056_

Round 1

Reviewer 1 Report

Thank you so much for allowing me to review this paper.

The authors indicated during the Covid-19 pandemic; mental health is reported to have worsened. A study was conducted to assess whether this occurred among 20 pregnant women who were unaware of the pandemic at the time of the survey (late 2019) and who delivered after the implementation of Covid-19 restrictions and threat 21 (March-April 2020). According to the authors, a retrospective 2014-2015 sample of 22 people with affective symptoms was used to compare the pandemic period with the pre-pandemic period.

I found some suggestions for this manuscript for significant revision.

Q1. Further extended some implications of the outcome from the abstract.

Q2. Suggested explaining more correlation between the COVID-19 pandemic on postpartum maternal mental health. 

Q3. Why choose the target of women adults? Any comparison differences? If yes, please explain in the introduction section and the literature review support. 

Q4. Please consist of the wording COVID-19 pandemic or COVID-19 epidemic. 

Q5. Rewrite the passage lines261-282.

Q6. Suggest adding the "Future implication" and the "limitation" section.

Q7. For the citation that develops more about COVID-19 and patient situations, I suggest adding the reference paper https://doi.org/10.3389/fmed.2021.666973.

Author Response

Response to Reviewer 1 Comments

Point 1. Further extended some implications of the outcome from the abstract.

Response 1: Thank you for your suggestion, we extended some implications of the outcome from the abstract.

Point 2. Suggested explaining more correlation between the COVID-19 pandemic on postpartum maternal mental health. 

Response 2: Thank you for your suggestion, we explained more correlation between the COVID-19 pandemic on postpartum maternal mental health in the discussion.

Point 3. Why choose the target of women adults? Any comparison differences? If yes, please explain in the introduction section and the literature review support. 

Response 3: Thank you for your question, we recruited women in their perinatal period in the context of a  screening programme for maternal mental health

Point 4. Please consist of the wording COVID-19 pandemic or COVID-19 epidemic. 

Response 4: Thank you, we have done

Point 5. Rewrite the passage lines261-282.

Response 5: Thank you, we have done

Point 6. Suggest adding the "Future implication" and the "limitation" section.

Response 6: Thanks for the suggestion, we've added the "Future Implications" and "Limitation" sections.

Point 7. For the citation that develops more about COVID-19 and patient situations, I suggest adding the reference paper https://doi.org/10.3389/fmed.2021.666973.

Response 7: Thank you, we have added the article you suggested

Reviewer 2 Report

Reviewer Report

This study investigates the impact of the COVID-19 pandemic on the mental status of pregnant or postpartum women; the study design utilizes pre-pandemic COVID-19 data and compares them to post-pandemic studies. The survey items are detailed, the statistical methods are straightforward, and the discussion from the survey results is grounded.

We believe that the results of this study provide beneficial direction for the maternal and child health field, both during the COVID-19 pandemic and in the future after the COVID-19 pandemic.

Major Comments

Except in extensive, global epidemiologic studies, regional bias is unavoidable. This study was conducted in a single region.

The description of regional bias is well stated in the discussion part (L351-). However, there is a lack of discussion regarding the relevance of other regions.

I think it is necessary to expand the generality of the discussion by using previous studies in other regions related to this study, if possible. Please consider this.

This is my personal view, so please refer to it if you like.

Regarding the Conclusions part, I think it should be simple, if possible. The primary purpose of this part should be to answer the aim of the study. Therefore, it would be better to describe the new findings of this study in a way that answers the aim of the study so that the excellence of this study can be clarified.

We hope you will consider this.

Please note that this is also a personal viewpoint.

Four critical tables have been included in this study. The study covers a wide range of topics and displays a great deal of valuable data. The tables, more dividing lines there are, the more difficult it will be to read the tables. We would appreciate it if you would reconsider how the tables are displayed.

Thank you in advance for your consideration.

Author Response

Point 1. The description of regional bias is well stated in the discussion part (L351-). However, there is a lack of discussion regarding the relevance of other regions. I think it is necessary to expand the generality of the discussion by using previous studies in other regions related to this study, if possible. Please consider this. This is my personal view, so please refer to it if you like.

Response 1: Thank you for your suggestion, we have reviewed the literature on studies in other regions as you can find in the discussion

Point 2. Regarding the Conclusions part, I think it should be simple, if possible. The primary purpose of this part should be to answer the aim of the study. Therefore, it would be better to describe the new findings of this study in a way that answers the aim of the study so that the excellence of this study can be clarified. We hope you will consider this.. 

Response 2: Thank you for your suggestion, we have simplified the conclusion as suggested

Point 3. Four critical tables have been included in this study. The study covers a wide range of topics and displays a great deal of valuable data. The tables, more dividing lines there are, the more difficult it will be to read the tables. We would appreciate it if you would reconsider how the tables are displayed.

Response 3: Thank you for acknowledging the serious data work. We are sorry that the tables are not easy to read. Hopefully the graphics can help with this inconvenience.